# Genomic Association between SNP Markers and Diseases in the “Curraleiro Pé-Duro” Cattle

**DOI:** 10.3390/genes12060806

**Published:** 2021-05-25

**Authors:** Thais Miranda Silva Freitas, Juliana Moraes Dias, Luanna Kim Pires Guimarães, Sáudio Vieira Peixoto, Rayanne Henrique Santana da Silva, Kareem Rady Badr, Maria Ivete Moura, Adriana Santana do Carmo, Vincenzo Landi, Maria Clorinda Soares Fioravanti

**Affiliations:** 1School of Veterinary and Animal Science, Federal University of Goiás, Campus Samambaia, Goiânia, Goiás 74690-900, Brazil; julianadias@ufg.br (J.M.D.); luannakim@hotmail.com (L.K.P.G.); saudiovet@hotmail.com (S.V.P.); rayanne.henrique@discente.ufg.br (R.H.S.d.S.); adrianacarmo@ufg.br (A.S.d.C.); clorinda@ufg.br (M.C.S.F.); 2Environmental Virology Laboratory, Water Pollution Research Department, National Research Centre, Dokki, Giza 12622, Egypt; kareem_nrc@yahoo.com; 3Course in Veterinary Medicine, Pontifical Catholic University of Goiás, Av. Engler, Jardim Mariliza, Goiânia, Goiás 74885-460, Brazil; mariaivete@pucgoais.edu.br; 4Department of Veterinary Medicine, University of Bari Aldo Moro, Str. Prov. per Casamassima, Km 3, 70010 Valenzano, BA, Italy; vincenzo.landi@uniba.it

**Keywords:** local breed, polymorphism, disease susceptibility

## Abstract

Susceptibility to diseases is inherited and can be transmitted between populations. Single-nucleotide polymorphism (SNPs) in genes related to immune response is associated with diseases in cattle. This study investigated SNPs in the genomic region of cytokines in 702 samples of Curraleiro Pé-Duro cattle and associated them with the occurrence of antibodies in brucellosis, leptospirosis, neosporosis, leukosis, infectious bovine rhinotracheitis (IBR), and bovine viral diarrhea (BVD) tests. DNA samples were evaluated by the kompetitive allele-specific polymerase chain reaction (KASP) method to identify polymorphisms. The gametic phase and SNP haplotypes were determined with the help of PHASE 2.1.1 software. Haplotypes were associated with serological results against *Brucella abortus*, *Leptospira* sp., *Neospora caninum*, leukosis, infectious rhinotracheitis, and BVD using univariate analysis followed by logistic regression. Haplotype 2 of *TLR2* was present in 70% of the animals that tested positive for *N. caninum* infection. Haplotypes of *TLR10* and *TLR6* and *IL10RA* were more common in seronegative animals. Haplotypes related to the gene *IL10RA* were associated with animals negative to all infections. Curraleiro Pé-Duro cattle presented polymorphisms related to resistance to bacterial, viral, and *N. caninum* infections.

## 1. Introduction

Curraleiro Pé-Duro (CPD) is a local Brazilian bovine breed with historical, cultural, and ecological values, but its population size is small and faces the risk of extinction [1]. The breed is rustic and shows high resistance to regional endemic diseases [2] because of the high level of circulating T-lymphocytes, which decreases with age. Besides, it has high levels of lymphocytes, which give the breed resistance to hemoparasitosis [3], *Mycobacterium bovis*, and other infections [2].

Disease resistance is determined by many genes, including the ones that encode the regulatory molecules of the immune system. One of the most critical genomic regions involved in disease resistance is the bovine histocompatibility complex, called bovine leukocyte antigen (*BOLA*), containing a set of closely linked polymorphic genes that code for cell surface proteins essential for the adaptive immune system [4]. The complex includes antigen-presenting genes, complementary system genes that attack the antigen, and genes related to the tumor necrosis factor (*TNF*) [5]. In the presence of a pathogen infection, the immune response of animals can respond differently, in accordance with the polymorphism in its antigen receptor genes [6].

The immune response depends of the contributions of multiple genes that produce molecules involved in the regulating the activity of the host immune system. The complex regulation of immunity includes the gene encoding protein 1 (*NRAMP 1*) [7], which is also known as solute carrier family 11 member 1 (*SLC11A1*), immunoglobulin genes, and T-cell receptor genes [8]. *SLC11A1* encodes an iron ion transmembrane transport protein, while the toll-like receptor 1 (*TLR1*) encoding gene is responsible for pathogen-associated molecular pattern-recognition proteins (*PAMP*s) and regulates cytokine production [9,10,11]. Cytokines are a group of proteins involved in the recruitment of inflammatory cells and allelic variations in genes and cytokine receptors that modify the phenotypic response to infection [12].

The type of pathogen involved in the infection determines the type of immune response. Toll receptors (toll-like receptor—*TLR*), which are present in phagocytic cells, are activated by microbial products and emit intracellular signals that activate cytokine production. The *TLR*2 receptor is activated by cell wall molecules of Gram-positive bacteria, *TLR*4 by bacterial membrane lipopolysaccharides, *TLR*5 by flagellin, and *TLR*9 by bacterial DNA [13,14]. The Toll gene family products participate in the recognition of bacteria, virus, fungi, and protozoa, inducing host innate immune responses [15].

Bacterial infections, such as those caused by *Brucella abortus*, activate the cellular immune response, generating cytokines that stimulate the bactericidal activity of macrophages and cytotoxicity of CD8 T-lymphocytes, which destroy infected cells. The production of interleukin 12 (*IL-12*) by antigen-presenting cells stimulates natural killer cells that lyse infected cells [13,16]. *N. caninum* infection triggers a cellular response mediated by pro-inflammatory cytokines such as interferon-γ (*IFN-γ*), tumor necrosis factor-α (*TNF-α*), and the induction of inducible nitric oxide synthase (*iNOS*) associated with phagocytic activity [17]. The proteins present on the outer surface of *Leptospira* sp. modulate the immune response mediated by *TLR*s and the production of pro-inflammatory cytokines [18].

Bovine leukosis virus (VLB), bovine herpesvirus type 1 (BoHV-1), and bovine viral diarrhea virus (BVDV) induce the production of *TNF-α*. In vitro BVDV infection, in the presence of another pathogen, decreases the production of *TNF-α*, thereby contributing to the immunosuppression commonly observed in animals infected postnatally. There is a decrease in chemotaxis and the release of an inhibitor of interleukin 1 activity, which enhance the survival of the virus in the host [19].

The persistent infection in the fetus infected by the bovine viral diarrhea virus (BVDV) noncytopathic (ncp) strain induces higher neutralizing antibodies as compared to homologous cytopathic strain [20] and establishes lifelong infections and immunosuppression. After ncpBVDV infection, *IFN-γ* is induced and promotes cytokine and T-cell response, interfering in immune response to pathogens, such as bovine herpesvirus 1 [21].

Given the genetic role in immune regulation, marker-assisted selection can be a cheap and simple tool to identify the resistant animals based on single nucleotide polymorphism (SNPs) variations involved in the response to pathogens [22]. It is important to find genetic variations associated with diseases [6], especially SNPs that are disseminated in candidate gene regions, whose effects are not yet very well defined [23].

Single nucleotide polymorphisms are a substitution of one nucleotide base in DNA sequence, some of which can affect gene transcription and protein activity, promoting differences between species and individuals. SNPs’ markers are used in genetic and genome-wide association studies, because they can be found throughout the genome and are stably transmitted to progeny. Most SNPs are biallelic and have lower information content in comparison to microsatellite markers, so many SNPs are necessary to evaluate differences in population [24,25].

A combination of alleles occurring on the same autosome or sexual chromosome region, inherited together in a population by the principle of linkage disequilibrium, is termed a haplotype. Each individual has two haplotypes, and one population may have numerous haplotypes. In regions of high linkage disequilibrium, haplotypes that contain, or are correlated to, casual variants can serve to identify disorders or diseases [26].

This paper aims to identify the SNPs’ haplotypes in immune response genes and their association with serology test results in Curraleiro Pé-Duro cattle raised in Brazil.

## 2. Materials and Methods

### 2.1. DNA Sampling and Extraction

The samples were obtained from the Pró-Centro Oeste Network database, research, and knowledge transfer in Brazil, which maintains blood, serum, and DNA samples as well as epidemiological data of CPD cattle herds. Aliquots of blood samples of animals of both sexes and all ages, contained in the database of samples of the Network at the Universidad de Córdoba, Spain, were subjected to DNA extraction by the salting out method [27] with some modifications (Appendix A). After extraction, the DNA was quantified in a NanoVue Plus^TM^ spectrophotometer (Biochrom, Holliston, MA, USA) and standardized at 20 ng/µL, excluding samples that presented OD260/OD280 lower than 1.5 and higher than 2.0.

The samples were separated; 940 samples corresponding to 10 plates with two controls in each were sent to the LGC Genomics laboratory (www.lgcgenomics.com (accessed on 23 February 2017)) in the United Kingdom, where they were processed by the kompetitive allele-specific polymerase chain reaction (KASP) technique.

All samples of the clinical specimens bank were collected in 2010–2011 by jugular venopunction under the welfare principles according the Ethics Committee for the Use of Animals (CEUA) of Federal University of Goiás, protocol Nº 106/19.

### 2.2. Genotyping and Estimation of Haplotypes

The SNPs used (Table 1) were present in the genic regions of integrin genes, *TLR*s, node-like receptors, interleukins, and interferon-γ and were related to tick infestations in cattle [28], subclinical mastitis [29], and bovine leukosis [30]. The primers were designed from DNA sequences available in the Ensembl online database for bovine species (http://www.ensembl.org/Bos_taurus/Info/Index (accessed on 20 June 2016)).

With the help of Kbioscience software, the wavelength data emitted by FAM and HEX were plotted on the x- and y-axes, respectively, and ROX values were used as a reference for data normalization, eliminating problems related to the variation in the volume of liquid in each sample. The homozygous genotypes were separated from heterozygous genotypes according to the marked color and position in the graph [31]. The genotyping assay was validated, and genotype determination was done using KlusterCaller 1.1 (LGC Genomics) software.

Quality control of genotype data was performed using the software Excell. Monomorphic SNPs were removed, and SNP markers with a minor-allele frequency (MAF) <1%, and individuals with a call rate <85% were discarded to preserve samples with the best quality of genotyping [32], leaving 702 SNP marker genotypes in the final dataset.

Given the genotypes, the gametic phase of the individuals was determined with the help of PHASE 2.1.1 software to establish the haplotypes. SNPs were grouped according to the chromosomes in which they were present to define the effect of the combination of SNPs on the phenotype. For each chromosome analyzed, the algorithm was applied five times and 100 iterations, a sampling interval, and 100 burn-in through the Markov chain were used [34,35]. Each individual was classified into two haplotypes, thereby generating 1404 haplotypes.

### 2.3. Obtaining Phenotypes and Epidemiological Variables

In Brazil, the cattle are mainly raised in pasture systems, which implies an exposure to several pathogenic agents simultaneously. Thus, this study was designed to identify haplotypes associated with resilience in local breeds populations adapted to harsh environments and naturally exposed to infections. For locally adapted breeds from small cattle populations, the limited number of records for quantitative traits, especially for fitness and health traits, is a special challenge in genomic studies.

Individuals from 19 different farms were selected for genotyping using a selective genotyping approach. In this regard, the selection criteria were the herd prevalence for the diseases considered in the study, which indicates the herd was naturally exposed to the pathogens.

The phenotypes were defined according to the positive or negative reaction to serological tests [9]. The analysis of association with the phenotypes was carried out through univariate analysis, followed by logistic regression, which had the serological result as the response variable and haplotype/polymorphism as the other variables.

The SNPs were associated with a serological reaction (positive or negative) against *B. abortus*, *Leptospira* sp., *N. caninum*, bovine leukosis virus, bovine herpesvirus type 1 (causing infectious bovine rhinotracheitis), and bovine viral diarrhea (BVD). Serological analyses are a part of the Pró-Centro Oeste Network database and were carried out from 2011 to 2013.

Serological tests for viral infections included enzyme immunoassay tests (IDEXX Laboratories, Inc. Westbrook, ME, USA), which were performed according to the manufacturer’s cut-off point. The buffered acidified antigen (AAT) method (Instituto de Tecnologia do Paraná—TECPAR^®^, Curitiba, Brazil) was used against *B. abortus*, considering the agglutinated samples positive, and the 1:100 cut-off point was used in microagglutination tests against 19 serovars of *Leptospira* sp. (collection of antigens from the Laboratory of Bacterial Zoonosis of the University of São Paulo) and indirect immunofluorescence against tachyzoites of *N. caninum* NC-1 isolate [9].

### 2.4. Statistical Analysis

Haplotypes on each chromosome were associated with the serological response. In chromosomes that had only one SNP, only the presence or absence of polymorphism was considered. Each individual has a pair of haplotypes, and hence, the serological response variables were duplicated so that each haplotype was computed only once.

A multivariate analysis was applied to identify markers with the largest effect on phenotypic characteristics. With the odds ratio value, it was possible to observe alleles related to susceptibility. Data were analyzed with the R (R Core Team, Vienna, Austria) statistical program using the packages Epitools for univariate analysis and Epicalc and Car for regression. A 5% significance level was considered in the Chi-square and Fisher’s tests [23,36].

The epidemiological variables obtained from the epidemiological questionnaires of the Pró-Centro Oeste Network were herd size, rearing of other breeds, animal acquisition, quarantine enforcement, use of common pastures or rental of pastures, presence of flooded area, slaughter at the property, occurrence of abortions, vaccination, type of management, presence of rodents, and veterinary assistance. These variables are associated with the risk of infection; hence, some animals, even in the presence of an environment favorable for infections, may not exhibit the infection. The significant variables were added to the multivariate generalized linear model using binary logistic regression. For this purpose, the glm function of the R software was utilized [36].

## 3. Results

In the quality control stage, two monomorphic SNPs, rs207532826 and rs42395524, were removed from the analyses. Based on the call rate criterion, 238 samples with >15% failures were excluded, and only 702 of the 940 initial samples were left. SNP quality control prevents newly pinned markers and samples with low informative content from generating false results in association tests.

The frequencies of seropositive reactions were: 0.85% (6/702) for *B. abortus*, 42.07% (292/694) for *Leptospira* spp., 36.87% (257/697) for *N. caninum*, 21.07% (146/693) for VLB, 64.65% (450/696) for BoHV-1, and 40.38% (279/691) for BVD. The frequencies of each infection by property (owner) were calculated by considering only the samples used in the evaluation of SNPs (Table 2).

The low prevalence of B. abortus agglutinins expresses the effective control of disease by the National Program for the Control and Eradication of Animal Brucellosis and Tuberculosis and the correct vaccination of heifers against brucellosis. On the contrary, there is no vaccine available against N. caninum infection in cattle, and the control strategies are not suitable.

To assess whether SNPs are in linkage disequilibrium, the markers were grouped according to the chromosome, and haplotypes representing different genotypes were formed (Table 3). SNP rs42395526 was significant in all infections.

The frequencies of infections were tested for each polymorphism by univariate analysis using Fisher’s test, and the significant differences are presented in Table 4. The analyses identified SNPs associated with infection in the 10 chromosomes that were studied.

Haplotypes related to the described infections were identified by the Chi-square analysis, considering *p* < 0.05 (Table 5).

The analyses were carried out in two steps: (1) identification of the epidemiological variables that represent risk factors for the onset of diseases through the Chi-square test. The non-significant variables were evaluated as having little relevance for increasing the incidence of the disease; (2) identification of haplotypes associated with diseases. Logistic regression was performed considering haplotypes and significant epidemiological variables as independent variables in the model. The final regression results are expressed in Table 6.

Logistic regression was used to observe whether the response variable (presence/absence of antibodies) was related to the haplotypes of each chromosome and related to epidemiological variables.

When the epidemiological factors were included into ANOVA analysis (Table 7), the contrasts evidenced that epidemiological factors have an important role in disease incidence.

Haplotypes c8_h1 and c15_h2 were present in five of six positive *B*. *abortus* animals. For *Neospora* sp. infections, we found c10_h7 haplotype in 25.29% (130/514) positive animals, c17_h2 haplotype present in nine negative and 21 positive animals. Haplotype c18_h1 present in 104 positive and 130 negative animals. In bovine leucosis infection, the haplotype c27_h2 was present in 31.5% (46/146) positive animals.

It was possible to identify potential gene loci candidates for bovine resistance to infectious diseases in the 10 chromosomes studied. The BTA15 chromosome haplotypes were significant against the infections defined. Of the five haplotypes described, haplotypes 1 and 2 (c15_h1 and c15_h2) were more frequent in seronegative animals. Haplotypes 1, 5, and 6 in BTA6 were more common in animals negative against leptospirosis, neosporosis, viral diarrhea, and leukosis.

Haplotype blocks containing two, three, and four SNPs were found that were significantly (*p* < 0.05) associated with positivity or negativity for the infection. Haplotypes were generally more frequent in seronegative animals and were, therefore, associated with the absence of infection.

## 4. Discussion

The high frequencies of *Leptospira* sp., *N. caninum*, IBR, and BVD in CPD herds can be attributed to these diseases being endemic. Therefore, most animals would have been exposed at some stage of life, thereby resulting in the production of antibodies.

The interaction between genetic haplotypes and the environment demonstrates that environmental variables were more important to define the risk to disease. Nevertheless, some haplotypes were significant despite the environment.

Polymorphisms in *TLR*6 and *TLR*10 were more frequent in animals negative for leptospirosis, neosporosis, viral diarrhea, and leukosis, suggesting that this chromosome has candidate genes for resistance to infections. BTA6 contains information to encode *TLR*10 receptors, cells that act on the innate response by *PAMP* recognition of pathogens and preventing tissue invasion. These genes are expressed in antigen-presenting cells (macrophages and dendritic cells) and in families capable of recognizing bacteria, protozoa, and fungi [37,38].

The pathogens investigated are sexually transmitted and are important causes of reproductive loss in cattle. In humans, the proteins expressed by *TLR*10 are found in the placental tissues and are involved in protecting the fetus. *TLR*10 functions as a co-receptor of *TLR*2, a peptidoglycan agonist of Gram-positive bacteria that acts by inducing cell apoptosis and a reduction in chemokine secretion [39]. Animals that correctly express *TLR*10 have a greater ability to fight infections and reduce the transmission of pathogens than those that do not express the receptor.

The SNPs examined in *IL10RA* were associated with all infections. The SNPs selected in BTA15 are present in the gene of interleukin 10 (*IL-10*) subunit α receptor. *IL-10* is a cytokine produced by CD4+ T-cells, B-cells, and macrophages [40]. *IL-10* has immunoregulatory potential and inhibits pro-inflammatory cytokines, mainly *FNT*, *IL-1*, and *IL-6*, produced by activated macrophages and monocytes [41]. Furthermore, *IL-10* reduces inflammation during infections, preventing the onset of deleterious lesions. However, it can also promote the persistence of pathogens by interfering with the immune response, such as the persistence of *Leptospira* spp. in the kidneys of reservoir animals [42].

The association between haplotypes in *IL10RA* and seronegative animals may be due to the regulatory effect of *IL-10*. In *Leptospira* spp. infection, *IL-10* increases expression in the early post-infection days in resistant rats, while pro-inflammatory cytokine levels decrease [43]. The immunoprotective role of *IL-10* during leptospirosis is to mitigate the deleterious effects caused by the increase in pro-inflammatory cytokines, such as interleukin 1 β (*IL-1β*), which is related to organ failure in infected hosts. On the other hand, it exerts an inhibitory effect on bacterial clearance, causing bacteria to remain in the kidneys and be eliminated by the host [44].

The X chromosome was associated with neosporosis from the Chi-square test but was not associated with any pathogen in the regression analysis. Gene-encoding *TLR*8 are found in this chromosome. *TLR*8 recognizes the invasion of viruses and induces the innate immune response; hence, it is often associated with BVD and type 3 parainfluenza caused by the same bovine herpesvirus type 1, resulting in infectious rhinotracheitis as well as other viral diseases. Because these diseases are disseminated, animals are likely to be exposed for a long time. Furthermore, owing to evolutionary pressure, the pathogen–environment interaction of ancestral populations may have generated patterns of variation within the *TLR*3 and *TLR*8 genes that are seen as selection signatures [37].

The SNP rs29026690, in the current study associated with BVD, was described in an investigation of the broad association of the genome with an increased proviral load of the bovine leukosis virus in Japanese Black herds (odds ratio 2.745), within a locus of independent quantitative characteristics (QTL—quantitative trait loci) on chromosome 23. In the same study, they identified a minor association of the disease with SNP rs17872126 (odds ratio 0.414) [30].

In the analysis of haplotypes associated with infections, significant differences were identified in chromosomes 5, 6, 8, 10, 15, 17, 18, 27, and X. Haplotypes from *IL10RA* and *TLR*3 were present in a high rate of seronegative animals and may indicate a possible genotype of resistance to bovine leukosis virus. On the other hand, 70% of animals positive for *N. caninum* infection had haplotype 2 from *TLR*2.

The herd and breed of animals were significant variables in a study of the association of paratuberculosis infection and SNPs in the region of the bovine *IFNG* gene, which encodes γ interferons playing an important role in the innate host response. In this study, alleles and haplotypes were significant only when not associated with other explanatory variables. The non-association between haplotypes and epidemiological characteristics in response to infection observed in this study does not denote the lack of association. The conclusions obtained from case-control epidemiological studies are limited, and differences in the susceptibility of animals could be due to several factors such as errors in the classification of individuals in the categories of case (positive) and control (negative) [23], as observed in some serological tests.

Even though other studies have shown the possibility of significant results between markers and characteristics, its occurrence in populations may not be replicable. If the same locus of quantitative characteristics is segregating into different populations, the results will still be different, because the allelic frequencies for each marker and the mutations are different. Genetic resilience can present various phenotypes, such as exposed individuals who do not develop an infection, exposed subjects who become asymptomatic carriers, and individuals who develop clinical signs of the disease but manage to cure themselves [9,45].

Alleles that confer resistance in one breed need to be validated in other breeds. It was observed that the alleles associated with a lower load of tick infestation in one population do not perform the same function in others. Genetic markers or haplotypes do not show precise effects between different breeds [28], indicating that finding a marker associated with a resistance or susceptibility characteristic that is useful in different populations and races is very difficult for any association study, regardless of whether it is aimed at tick infestation or infection by viruses, bacteria, parasites, or fungi.

Animals of local breeds have proven resistance to infections; however, the selection for resistance characteristics is still at an early stage. CPD animals can be immunologically challenged to investigate the type and magnitude of the immune response in those carrying the haplotypes identified as resistance markers. Animals carrying resistance alleles must be included in breeding programs, and their genetic material should be preserved in the germplasm bank.

It is known that cattle production characteristics have been widely researched owing to the direct economic return. However, investments in health and the development of biotechnology make it feasible to control diseases via molecular markers as a way of supporting the sanitary strategies that have already been employed.

## 5. Conclusions

SNP-type markers related to the risk of infectious diseases were present in animals of the CPD breed. The polymorphisms formed haplotype blocks for each chromosome, and chromosomes 5, 6, 8, 10, 15, 17, 18, 27, and X were associated with diseases. Haplotype 2 of *TLR*2 was present in 70% of animals positive for *N. caninum* infection, while haplotypes related to *IL-10* and *TLR*10 were common in seronegative animals. The epidemiological variables demonstrate significance with serological status. Further research in experimentally infected CPD animals in a controlled environment should be done to validate the association between the presented SNPs and diseases antibody levels.

## Figures and Tables

**Table 1 genes-12-00806-t001:** List of SNP markers tested in Curraleiro Pé-Duro samples.

dSNPs Accession	Var	Location of the Gene	Position	Alleles	IUPAC	BTA	MAF (%)
rs41594962	I	*integrin subunit α 11*	15137028	C/T	Y	10	0.320
rs42395522	S	*interleukin 10 subun. α receptor*	29101724	A/G	R	15	0.494
rs8193069	M	*toll-like receptor 4*	108838685	C/T	Y	8	0.196
rs42395525	S	*interleukin 10 subun. α receptor*	29102009	C/T	Y	15	0.347
rs17872126	I	*bos taurus proline rich 3 (prr3), mrna*	28223274	C/T	Y	23	0.396
rs55617272	M	*toll-like receptor 3*	15240722	G/A	R	27	0.104
rs55617351	NC	*toll-like receptor 8*	141005664	T/C	Y	X	0.333
rs42395524	S	*interleukin 10 subun. α receptor*	29101838	T/T	Y	15	monomorphic
rs42395526	S	*interleukin 10 subun. α receptor*	29102042	G/A	R	15	0.494
rs42852439	M	*toll-like receptor 3*	15241437	T/G	K	27	0.372
rs43710288	M	*nod like receptor 2*	19210671	A/T	W	18	0.168
rs29026690	I	*lymphocyte antigen 6 family member*	27421348	C/T	Y	23	0.077
rs207532826	NC	*toll-like receptor 8*	141004386	T/T	K	X	monomorphic
rs43616884	D	*fem-1 homologg b*	15094573	G/A	R	10	0.309
rs110491977	M	*toll-like receptor 2*	3952585	C/T	Y	17	0.021
rs55617286	M	*toll-like receptor 10*	59672820	C/G	S	6	0.150
rs43702941	M	*toll-like receptor 6*	59706074	C/T	Y	6	0.319
rs55617325	M	*toll-like receptor 10*	59672512	T/A	W	6	0.285
rs108954324	M	*interferon γ*	45830291	G/T	K	5	0.080
rs43710290	3U	*nod-like receptor 2*	19212600	C/T	Y	18	0.060
rs29025980	I	*fem-1 homologg b*	15082638	G/A	R	10	0.222
rs43706433	M	*toll-like receptor 2*	3952556	C/T	Y	17	0.365
rs55617437	M	*toll-like receptor 10*	59673169	C/T	Y	6	0.209
rs108949553	I	*integrin subunit α 11*	15153881	G/A	R	10	0.334

dSNPs accession: Ensembl database reference nu181
mber (http://www.ensembl.org (accessed on 20 April 2018)); Var: vaiant; I: intron variant; S: synonymous variant; M: missense variant; NC: non-coding transcript exon variant; D: downstream gene variant; 3U: 3 prime UTR variant; IUPAC: IUPAC SNP code; BTA: *Bos taurus* autosome; MAF: lower allelic frequency. Source: Based on Vázquez et al. [33] and Porto Neto et al. [28].

**Table 2 genes-12-00806-t002:** Frequency of infections in samples evaluated by SNP markers from Curraleiro Pé-Duro cattle herds.

Prop.	*Brucella*% (*n*)	*Leptospira*% (*n*)	*Neospora*% (*n*)	VLB% (*n*)	IBR% (*n*)	BVD% (*n*)	Total
G1	0 (0)	50.7 (36)	15.49 (11)	19.72 (14)	50.7 (36)	35.21 (25)	71
G2	0 (0)	56 (14)	8 (2)	0 (0)	40 (10)	32 (8)	25
G3	0 (0)	29.55 (13)	52.27 (23)	43.18 (19)	52.27 (23)	11.36 (5)	44
G4	1.52 (1)	56.06 (37)	10.61 (7)	3.03 (2)	81.82 (54)	75.76 (50)	66
G5	1.56 (1)	10.94 (7)	70.31 (45)	7.81 (5)	46.88 (30)	14.06 (9)	64
G6	10.53 (2)	47.37 (9)	68.42 (13)	21.05 (4)	84.21 (16)	26.32 (5)	19
G8	2.99 (2)	59.7 (40)	32.84 (22)	20.9 (14)	70.15 (47)	22.39 (15)	67
P1	0 (0)	27.27 (24)	62.5 (55)	2.27 (2)	56.82 (50)	51.14 (45)	88
P2	0 (0)	41.18 (7)	88.24 (15)	5.88 (1)	82.35 (14)	52.94 (9)	17
P3	0 (0)	83.33 (15)	100 (18)	33.33 (6)	66.67 (12)	50 (9)	18
P4	0 (0)	50.00 (9)	5.56 (1)	16.67 (3)	94.44 (17)	16.67 (3)	18
P5	0 (0)	23.68 (9)	7.89 (3)	65.79 (25)	78.95 (30)	42.11 (16)	38
P7	0 (0)	4.08 (2)	22.45 (11)	8.16 (4)	63.27 (31)	30.61 (15)	49
P8	0 (0)	100 (5)	40 (2)	0 (0)	60 (3)	0 (0)	5
P9	0 (0)	100 (13)	84.62 (11)	30.77 (4)	61.54 (8)	76.92 (10)	13
T1	0 (0)	80.7 (46)	28.07 (16)	40.35 (23)	61.4 (35)	50.88 (29)	57
T2	0 (0)	11.11 (3)	3.7 (1)	40.74 (11)	96.3 (26)	74.07 (20)	27
T3	0 (0)	16.67 (1)	16.67 (1)	0 (0)	16.67 (1)	0 (0)	6
T4	0 (0)	20 (2)	0 (0)	90 (9)	70 (7)	60 (6)	10

Prop.: property; %: percentage of positive animals; *n*: number of positive animals; Total: total number of animals tested per property.

**Table 3 genes-12-00806-t003:** Haplotypes and SNPs related to the immune response in Curraleiro Pé-Duro cattle.

BTA	Polymorphism	Position	Gene	Haplotype or SNP	Genotype or Allele	Nucleotides
5	rs108954324	45830291	*IFNG*	1 2	1 0	T G
6	rs55617325 rs55617286 rs55617437 rs43702941	59672512 59672820 59673169 59706074	*TLR10 TLR10 TLR10 TLR6*	1 2 3 4 5 6 7	0110 0100 0101 0001 1100 1101 1001	CGAC CGTC CGTT CCTT TGTC TGTT TCTT
8	rs8193069	108838685	*TLR4*	1 2	0 1	C T
10	rs29025980 rs43616884 rs41594962 rs108949553	15082638 15094573 15137028 15153881	*FEM1B FEM1B ITGA11 ITGA11*	1 2 3 4 5 6 7	0101 0001 0000 0010 1111 1000 1010	GTGA GCGA GCGG GCAG ATAA ACGG ACAG
15	rs42395522 rs42395525 rs42395526	29101724 29102009 29102042	*IL10RA* *IL10RA* *IL10RA*	1 2 3 4 5	110 111 100 010 000	GTG GTA GCG ATG ACG
17	rs43706433 rs110491977	3952556 3952585	*TLR2* *TLR2*	1 2 3	00 01 10	CC CT TC
18	rs43710288 rs43710290	19210671 19212600	*NLR2* *NLR2*	1 2 3	10 00 01	TC AC AT
23	rs29026690 rs17872126	27421348 28223274	*LY6G6F* *PRR3*	1 2 3 4	00 01 10 11	CC CT TC TT
27	rs55617272 rs42852439	15240722 15241437	*TLR3* *TLR3*	1 2 3	00 10 11	TG GG GA
X	rs55617351	141005664	*TLR8*	1 2	1 0	C T

**Table 4 genes-12-00806-t004:** Results of univariate analysis between SNPs and serology in Curraleiro Pé-Duro cattle.

Pathogen	SNP	BTA	Position	*p* (>Chi-sq)
*Brucella*	rs8193069	8	108838685	0.007
	rs42395526	15	29102042	0.002
*Leptospira*	rs43706433	17	3952556	0.000
	rs43702941	6	59706074	0.000
	rs55617325	6	59672512	0.002
	rs55617437	6	59673169	0.006
	rs42395525	15	29102009	0.001
	rs43710290	18	19212600	0.008
	rs42395526	15	29102042	0.002
*Neospora*	rs110491977	17	3952585	0.009
	rs43702941	6	59706074	0.037
	rs42395522	15	29101724	0.002
	rs55617351	X	141005664	0.000
	rs42852439	27	15241437	0.014
	rs55617272	27	15240722	0.000
	rs43710288	18	19210671	0.009
	rs108954324	5	45830291	0.022
	rs43710290	18	19212600	0.024
	rs29025980	10	15082638	0.000
	rs42395525	15	29102009	0.023
	rs42395526	15	29102042	0.000
	rs43616884	10	15094573	0.006
VLB	rs43702941	6	59706074	0.031
	rs42395526	15	29102042	0.010
BoHV-1	rs43710288	18	19210671	0.039
	rs42395525	15	29102009	0.029
	rs42395526	15	29102042	0.027
BVDV	rs110491977	17	3952585	0.013
	rs43706433	17	3952556	0.000
	rs8193069	8	108838685	0.020
	rs43702941	6	59706074	0.001
	rs55617272	27	15240722	0.015
	rs41594962	10	15137028	0.000
	rs43710290	18	19212600	0.047
	rs29025980	10	15082638	0.043
	rs17872126	23	28223274	0.042
	rs42395526	15	29102042	0.005

BTA: Bos taurus autosome.

**Table 5 genes-12-00806-t005:** Significant Chi-square test results of SNP haplotypes and infections in Curraleiro Pé-Duro cattle.

Pathogen	BTA	Haplotype	*p*-Value
*Brucella*	8	1 and 2	<0.01
	15	2	<0.05
	27	3	<0.05
*Leptospira*	5	1 and 2	<0.05
	6	1 and 5	<0.05
	8	1 and 2	<0.05
	10	5	<0.05
	15	2	<0.05
	17	1 and 3	<0.01
*Neospora*	6	6	<0.05
	10	2 and 7	<0.05
	15	1, 2, and 5	<0.01
	17	1 and 2	<0.01
	18	1 and 3	<0.05
	27	1 and 3	<0.01
	X	1 and 2	<0.01
*Leucosis*	6	4 and 5	<0.05
	15	1 and 2	<0.01
	27	2 and 3	<0.05
IBR	15	2 and 3	<0.01
	18	1	<0.05
BVD	6	5	<0.01
	15	2	<0.01
	17	1 and 3	<0.01
	18	1	<0.05
	23	2	<0.05
	27	2	<0.05

**Table 6 genes-12-00806-t006:** Final model of logistic regression of SNP haplotypes and infectious diseases in Curraleiro Pé-Duro cattle.

Pathogen	BTA	Hp	Negative (*n*)		Positive (*n*)		Suspect (*n*)	*p*	Odds	95% CI
*Brucella*	8	1	99.56%	(1132/1137)	0.43%	(5/1137)	0% (0/0)	0.009	0.202	0.06–0.67
	15	2	98.03%	(349/356)	1.96%	(7/356)	0% (0/0)	0.015	4	1.30–13.4
*Leptospira*	6	1	49.87%	(200/401)	48.37%	(194/401)	0% (0/0)	<0.01	1.588	1.25–2.01
	10	5	31.57%	(6/19)	68.42%	(13/19)	0% (0/0)	0.012	3.298	1.24–8.77
	15	2	62.64%	(223/356)	36.79%	(131/356)	0% (0/0)	<0.01	0.717	0.55–0.92
*Neospora*	6	6	82.85%	(29/35)	17.14%	(6/35)	0% (0/0)	<0.01	0.325	0.13–0.80
	10	2	3.33%	(1/30)	56.66%	(17/30)	0% (0/0)	0.013	2.605	1.21–5.56
	10	7	54.82%	(159/290)	44.82%	(130/290)	0% (0/0)	<0.01	1.486	1.12–1.98
	15	1	75.00%	(96/128)	24.21%	(31/128)	0% (0/0)	<0.01	0.424	0.27–0.66
	15	5	66.85%	(476/712)	32.16%	(229/712)	0% (0/0)	<0.01	0.528	0.41–0.67
	17	2	30.00%	(9/30)	70.00%	(21/30)	0% (0/0)	<0.01	3.842	1.68–8.76
	18	3	75.90%	(63/83)	22.89%	(19/83)	0% (0/0)	0.021	0.54	0.31–0.93
	18	1	55.08%	(130/236)	44.06%	(104/236)	0% (0/0)	<0.01	1.529	1.12–2.07
	27	1	66.62%	(587/881)	32.34%	(285/881)	0% (0/0)	<0.01	0.64	0.49–0.85
	27	3	48.63%	(71/146)	51.36%	(75/146)	0% (0/0)	0.018	1.61	1.08–2.41
VLB	6	5	72.31%	(222/307)	26.05%	(80/307)	0.3% (1)	0.020	1.43	1.06–1.92
	15	1	67.18%	(86/128)	32.81%	(42/128)	0% (0)	0.038	1.546	1.03–2.31
	15	2	81.46%	(290/356)	15.16%	(54/356)	0.2% (1)	<0.01	0.616	0.44–0.86
	27	3	85.61%	(125/146)	13.69%	(20/146)	0% (0)	0.015	0.56	0.34–0.92
BVDV	6	5	58.30%	(179/307)	31.92%	(98/307)	7.2% (22)	<0.01	0.669	0.51–0.87
	15	2	57.58%	(205/356)	33.14%	(118/356)	8.1% (29)	0.013	0.731	0.57–0.93
	23	2	63.63%	(56/88)	26.13%	(23/88)	10.2% (9)	0.030	0.611	0.38–0.96
	27	2	45.35%	(171/377)	45.09%	(170/377)	8.7% (33)	0.015	1.347	1.05–1.71
BoHV-1	15	2	38.48%	(137/356)	60.39%	(215/356)	0.8% (3)	0.028	0.744	0.57–0.96

**Table 7 genes-12-00806-t007:** Variables included in the logistic regression model and Bonferroni adjust p-value for disease status and epidemiological and SNP haplotype variables in Curraleiro Pé-Duro cattle.

Pathogen	Variable	Odds Ratio	*p*-Value of ANOVA Test	*p*-Value Adjust by Bonferroni
*Brucellla*	c8_h1	0.1787	0.0013	0.0048 × 10^−1^
	c15_h2	4.9529	0.0019	0.0083
	No brucellosis vaccine	-	0.0134	0.06
*Leptospira*	Only one breed	0.0430	2.17 × 10^−12^	4.2 × 10^−6^
	No flooded area	0.0594	<2.2 × 10^−16^	1.1 × 10^−8^
	No abortion	4.4 × 10^8^	6.10 × 10^−10^	1.9 × 10^−5^
	No leptospira vaccine	4.2774	0.0116	<2 × 10^−16^
	No brucellosis vaccine	2.0939	0.0241	9.4 × 10^−6^
	No presence of rodents	32.8446	3.94 × 10^−10^	5.6 × 10^−8^
*Neospora*	c10_h7	1.5233	0.0006	0.0013
	c17_h2	1.1390	0.03	0.0014 × 10^−1^
	c18_h1	1.5397	0.1317 × 10^−4^	0.0085
	No flooded area	0.1320	3.037 × 10^−14^	2.7 × 10^−12^
	No leptospira vaccine	0.0167	0.0038	0.042
	No presence of rodents	0.0184	1.207 × 10^−12^	6.0 × 10^−12^
	No veterinary assistance	0.2543	<2.2 × 10^−16^	<2 × 10^−16^
Leukosis	c27_h2	1.7692	0.0019	0.063
	Only one breed	0.1337	6.99 × 10^−7^	1.0 × 10^−1^
	No quarantine	4.0114	0.2749	3.5 × 10^−6^
	No abortion	-	3.76 × 10^−12^	0.12
	No leptospira vaccine	-	8.62 × 10^−11^	1.5 × 10^−7^
	No brucellosis vaccine	1.7503	0.5709	4.5 × 10^−5^
	No presence of rodents	-	0.0007	3.3 × 10^−11^
	No veterinary assistance	4.1974	3.99 × 10^−10^	<2 × 10^−16^
IBR	No quarantine	2.4422	0.0006	5.3 × 10^−5^ (positive × not applicable)
	No on-farm slaughter	0.1467	5.06 × 10^−15^	4.3 × 10^−12^
	No leptospira vaccine	2.5933	5.18 × 10^−5^	0.57
BVD	No quarantine	0.3711	<2.2 × 10^−16^	<2 × 10^−16^
	No on-farm slaughter	0.1848	5.87 × 10^−16^	<2 × 10^−16^
	No leptospira vaccine	10.8361	1.15 × 10^−14^	0.01
	No presence of rodents	5.8510	9.74 × 10^−8^	1.6 × 10^−7^

## Data Availability

The data presented in this study are available in figshare. Dataset. doi:10.6084/m9.figshare.14130623.

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
