# Peer review of "Genomic Association between SNP Markers and Diseases in the “Curraleiro Pé-Duro” Cattle"

_genes, 2021, doi:10.3390/genes12060806_

Round 1
Reviewer 1 Report
Authors in the manuscript evaluated genomic association/ the polymorphism (SNP markers) in Curraleiro Pé-Duro cattle with disease resistance and immune response. Author evaluated the SNPs in 702 sample (after excluding 238 samples from a total 940 samples) and concluded that SNP-type markers is directly related to the risk of infectious diseases in CPD cattle. Polymorphisms in BTA6 were more frequent in animals negative for leptospirosis, neosporosis, viral diarrhea, and leukosis, similarly haplotypes of BTA6 and BTA15 were more common in seronegative animals while Haplotype 2 of BTA17 was associated with N. caninum positive animals. Although authors did meticulous work in conducting the experiments but experimental design and manuscripts has lot of issue to be fixed. Which are as follows
Major comments
- There was no proper control in the study. Authors determined the SNPs and compare that results with serological results. There are chances that seronegative animals will be more resistant (produce more antibody) to the disease once they are exposure the infectious agent without change in SNPs. To precisely determine relationship between SNPs with seroconversion. Seronegative animals need to be exposed to infection and then measure the effect on SNPs. If it is not possible, author could explore the availability for samples (Previous study which has proper control) and confirm their results with that.
- Authors in the abstract mentioned about polymorphism in bovine leukocyte antigen (BoLA). However it was not mention anywhere in the manuscript (BoLA-DRB genes). However author mentioned other SNPs like in gene for interleukin, integrin, TLRs etc.
- Authors nicely explained about the gene for immunological importance and immune response/infection in the introduction. However, Authors did not mention (provide any background) what is Single nucleotide polymorphisms (SNPs)/polymorphism, Bos Taurus autosomes (BTA), haplotype and their significance in disease resistance or immunity. However, the manuscript is focused on that.
Minor comments.
- Page 3, line 103. It is mentioned that total number of samples were 940 while page 5, line 177 it is 936.
- Page 2, line 80. Reference is missing in support of various pathogen induce TNF alpha.
- Page 2 line 81-84. There are many/more recent studies that only non-cytopathic strain of BVD cause immune suppression and persistent infection not the cytopathic strain.
Author Response
Major comments
- There was no proper control in the study. Authors determined the SNPs and compare that resultswith serological results. There are chances that seronegative animals will be more resistant (produce more antibody) to the disease once they are exposure the infectious agent without change in SNPs. To precisely determine relationship between SNPs with seroconversion. Seronegative animals need to be exposed to infection and then measure the effect on SNPs. If it is not possible, author could explore the availability for samples (Previous study which has proper control) and confirm their results with that.
Thanks for the note. We partially agree or rather there is the possibility that there are animals resistant not only to the infection but also to the entry of the pathogen to be seronegative. In our study we considered many animals in very different habitats and all potentially exposed to pathogens to the same extent. By comparing the markers, we also aim to measure the resilience of the animals, i.e., the ability to come into contact with the pathogen and not be influenced by it. In fact, in a second part of our study we are carrying out experimental infections on a group of animals to precisely verify this possibility.
- Authors in the abstract mentioned about polymorphism in bovine leukocyte antigen (BoLA). Howeverit was not mention anywhere in the manuscript (BoLA-DRB genes). However author mentioned other SNPs like in gene for interleukin, integrin, TLRs etc.
We apologize for the mistake. The first version of the manuscript contained the SNPs present in the region of the BoLA-DRB genes but it was removed, however the abstract was not edited. We have already made the adjustments in the text.
- Authors nicely explained about the gene for immunological importance and immune response/infection in the introduction. However, Authors did not mention (provide any background) what is Single nucleotide polymorphisms (SNPs)/polymorphism, Bos Taurus autosomes (BTA), haplotype and their significance in disease resistance or immunity. However, the manuscriptis focused on that.
Thank you for the correction. We add a new and more detailed description about this advice from line 95 to 107 of the revised version.
Minor comments.
- Page 3, line 103. It is mentioned that total number of samples were 940 while page 5, line 177 it is 936.
Corrected
- Page 2, line 80. Reference is missing in support of various pathogen induce TNF alpha.
Thank you for the observation. We add the citation to the following document that extensively report this mechanism Nahed Ismail et al 2006 (doi:10.1128/IAI.74.3.1846–1856.2006)
- Page 2 line 81-84. There are many/more recent studies that only non-cytopathic strain of BVD cause immune suppression and persistent infection not the cytopathic strain.
Thank you for the observation. We add the citation of the noncitypathic strain reported in Rajput et al. (2020) (h ttps://doi.org/10.1016/j.rvsc.2020.01.012) and Charleston et al. (2002) (doi: 10.1128/JVI.76.2.923-927.2002).

Reviewer 2 Report
In general, the identification of genetic variation associated with disease resistance is important to improving the overall resistance of our populations to disease. However, I have a number of concerns about this manuscript.
While the English language in this manuscript is fine, there are a number of pools worded statements. I have highlighted a few as examples but a thorough revision is needed.
Specific examples of poorly worded statements:
Line 41- the breed is rough?
Line 41- immense is not a precise enough adjective
Line 43, - not appropriate to start his sentence with Besides
Line 50-52 Sentence started with The variable forms - is unclear what you are attempting to convey here
Line 54 - but on the products of various genes is unclear
In the introduction as a whole the specific objectives of the research are unclear.
LIne 96-97 - an ethics statement regarding the collection of samples is vital for publication
Line 102-104- an indication of DNA sample quality should be included
Section 2.2- an explanation of why this method was chosen over a more traditional GWAS type case-control analysis is needed
Line 123-125- more detail on SNP filtering and data quality control is needed.
Section 2.3 More detail is needed here on how the different analyses were conducted. In addition, it appears that a multiple comparison adjustment should have been applied and wasn't. Please justify not utilizing one.
Section 3- Results
My interpretation is that when covariates were included in the models not significant associations were found and thus they were removed. This doesn't appear defensible to me. Please better explain why this was done and defend its validity.
With the frequency of seropositive appear to be very imbalanced especially for B. abortus and N. caninum, please defend the validity of those analyses as it does not appear to have enough power to made the comparisons.
Again in line 224-227 it is stated that logistic regression was related to haplotypes of each chromosome and infections with the epidemiological variables and therefore the risk factors were not significantly associated with the haplotype. Justify further analysis here as the epidemiological variables appear to be far important than the haplotype?
Considering the statement above, the discussion section is highly speculative and should be reframed to better acknowledge that last of significance stated above.
The conclusions do not seem supported by the data reported.
Author Response
RESPONSE TO REVIEWER 2
While the English language in this manuscript is fine, there are a number of pools worded statements. I have highlighted a few as examples but a thorough revision is needed.
Specific examples of poorly worded statements:
Line 41- the breed is rough? à Corrected
Line 41- immense is not a precise enough adjective à Corrected
Line 43, - not appropriate to start his sentence with Besides à Corrected
Line 50-52 Sentence started with The variable forms - is unclear what you are attempting to convey here à Corrected
The previous sentence: The variable forms in which animals respond to infection depend on mutations in the genes that modify the level of transcription and also on polymorphisms in antigen recognition sites (4).
The new sentence: In presence of a pathogen infection, the immune response of animals can respond of different forms, in accordance with the mutations and polymorphisms in its antigen receptor genes (4).
Line 54 - but on the products of various genes is unclear à Corrected
In the introduction as a whole the specific objectives of the research are unclear.
Thank you for the observation. We correct the objectives and the new sentence is:
This paper aims to identify the SNPs haplotypes in immune response genes and their association with serology test results in Curraleiro Pé-Duro cattle raised in Brazil.
LIne 96-97 - an ethics statement regarding the collection of samples is vital for publication
All samples of the clinical specimens bank were collected in 2010-2011 by jugular venopunction under the welfare principles according the Ethics Committee for the Use of Animals (CEUA) of Federal University of Goiás, protocol Nº 106/19
Line 102-104- an indication of DNA sample quality should be included à Corrected
After extraction, the DNA was quantified in a NanoPhotometerTM spectrophotometer and standardized at 20 ng/µL, excluding samples that presented OD260/OD280 lower than 1,5 and higher than 2,0.
Section 2.2- an explanation of why this method was chosen over a more traditional GWAS type case-control analysis is needed.
A brief explanation was added to the text.
In Brazil, the cattle are mainly raised in pasture systems which implies in exposure to several pathogenic agents simultaneously. Thus, this study was designed to identify haplotypes associated with resistance/resilience in local breeds populations adapted to harsh environments and naturally exposed to infections. For locally adapted breeds from small cattle populations, the limited number of records for quantitative traits, especially for fitness and health traits, is a special challenge in genomic studies1. Individuals from xxx different farms were selected for genotyping using a selective genotyping approach. In this regard, the selection criteria were the herd prevalence for the diseases considered in the study which indicates the herd was naturally exposed to the pathogens.
Line 123-125- more detail on SNP filtering and data quality control is needed.
More information was added.
Section 2.3 More detail is needed here on how the different analyses were conducted. In addition, it appears that a multiple comparison adjustment should have been applied and wasn't. Please justify not utilizing one.
We evaluated the presence/ausence of the disease associated to haplotypes and epidemiological variables on chi-square test and the results with p<0,05 were used in glm model.
We adjust the statistic model, we use the Bonferroni test to adjust the p value of Anova.
Section 3- Results
My interpretation is that when covariates were included in the models not significant associations were found and thus they were removed. This doesn't appear defensible to me. Please better explain why this was done and defend its validity.
The analyzes were carried out in two steps: 1) identification of the epidemiological variables that represent risk factors for the onset of diseases through the chi-square test. The non-significant variables were evaluated as having little relevance for increasing the incidence of the disease; 2) identification of haplotypes associated with diseases. Logistic regression was performed considering haplotypes and significant epidemiological variables as independent variables in the model.
With the frequency of seropositive appear to be very imbalanced especially for B. abortus and N. caninum, please defend the validity of those analyses as it does not appear to have enough power to made the comparisons. à Corrected.
The low-prevalence of Brucella spp. agglutinins express the effective control of disease by the National Program for the Control and Eradication of Animal Brucellosis and Tuberculosis, and the correct vaccination of heifers against brucellosis. In opposite, there is no vaccine available against Neospora caninum infection in cattle and the control strategies are not suitable.
Again in line 224-227 it is stated that logistic regression was related to haplotypes of each chromosome and infections with the epidemiological variables and therefore the risk factors were not significantly associated with the haplotype. Justify further analysis here as the epidemiological variables appear to be far important than the haplotype?
The text has been rewritten for better understanding. The haplotype and the epidemiological variables were evaluated as risk factors for the incidence of the diseases in glm model. The association between epidemiological variables and haplotypes was not tested.
.
Considering the statement above, the discussion section is highly speculative and should be reframed to better acknowledge that last of significance stated above.
The conclusions do not seem supported by the data reported.
After the adjustment proposed in statistic, we present the data for appreciation.

Round 2
Reviewer 1 Report
Authors did all the edits and included important informations in the revised manuscript. Revised manuscript is in good shape to publish
Author Response
Dear reviewers,
Thank you very much for the effort in evaluating our manuscript. We add the information proposed by the Academic Editor about the SNPs variation in Table 1 and correct the tables names. All the proposed corrections were done in the text and tables. We chance the disease resistance by disease susceptibility and the results are presented by SNPs and not by chromosome.
Reviewer 2 Report
My concerns have been addressed and this manuscript has been improved.
Author Response

(The authors gave the same response as above.)
